# Associations between the red blood cell distribution width-to-albumin ratio and 3-month outcomes in patients with acute minor ischemic stroke: A cohort study

Qin Chen[1], Qin Xiong[2], Li Liu[1], Lei Xu[1]*

1 Department of Neurology, Suining Central Hospital, Suining, Sichuan, China, 2 Department of Intensive Care Unit, the Third People's Hospital of Suining, Suining, Sichuan, China

* xulei@sns120.cn

## Abstract

Acute minor ischemic stroke (AMIS) patients may experience unfavorable outcomes despite mild initial symptoms. The ratio of red blood cell distribution width (RDW) to albumin (ALB) (RAR) has been associated with poor outcomes after stroke, but its role in AMIS remains unclear. This study aimed to investigate the association between the RAR and 3-month functional prognosis in patients with AMIS. This is a secondary analysis of a single-center, prospective cohort study conducted in Korea that included 988 patients with AMIS with a National Institutes of Health Stroke Scale (NIHSS) score ≤3. The RAR was calculated using RDW and ALB levels measured within 24 hours of admission. The functional outcome at 3 months poststroke was assessed using the modified Rankin Scale (mRS). The associations between the RAR and poor 3-month outcomes (mRS score ≥3) were evaluated using a multivariable logistic regression model, adjusting for confounders and conducting stratified and sensitivity analyses. Multivariate regression analysis revealed that the RAR was significantly associated with poor prognosis, both when it was unadjusted (OR = 2.29, 95% CI: 1.74–3.02, P < 0.0001) and after adjusting for age and sex (OR = 2.18, 95% CI: 1.63–2.91, P < 0.0001) and for multiple confounders (OR = 2.44, 95% CI: 1.67–3.57, P < 0.0001). Subgroup analysis revealed that the association between the RAR and poor prognosis was consistent across all subgroups. Elevated RAR is significantly associated with poor 3-month functional outcomes in patients with AMIS.

## Introduction

Stroke remains a leading cause of disability and mortality worldwide and poses a significant burden on healthcare systems and society [1]. Acute ischemic stroke (AIS), the most common type of stroke, is characterized by the sudden interruption of blood flow to the brain, leading to neurological deficits [2]. Among patients with

**Data availability statement:** All relevant data are within the paper and its Supporting Information files

**Funding:** The author(s) received no specific funding for this work.

**Competing interests:** The authors have declared that no competing interests exist.

AIS, those with acute minor ischemic stroke (AMIS), defined by a National Institutes of Health Stroke Scale (NIHSS) score ≤3, often present with subtle symptoms, which can lead to delayed diagnosis and treatment [3]. Despite the seemingly mild initial presentation, a substantial proportion of patients with AMIS experience poor functional outcomes, including disability and reduced quality of life, highlighting the need for accurate risk stratification and targeted interventions [4,5].

Red blood cell distribution width (RDW), an indicator of variability in the size of circulating red blood cells, and the level of albumin (ALB), a major plasma protein reflecting nutritional status and systemic inflammation, have been confirmed by numerous studies to be associated with adverse prognoses in cardiovascular disease and stroke [6–10]. RDW reflects the degree of red blood cell size heterogeneity and may be influenced by factors such as inflammation, oxidative stress, and erythropoiesis [11–13]. ALB, which is synthesized in the liver, is not only a marker of nutritional status but also an indicator of systemic inflammation and oxidative stress [14,15]. The ratio of RDW to ALB (RAR) has been proposed as a novel integrated marker that may offer additional prognostic value beyond individual markers alone [16].

Previous studies have demonstrated that elevated RDWs and decreased ALB levels are significantly associated with an increased risk of adverse poststroke outcomes [17–19]. Recent research has shifted focus to the synergistic predictive value of these biomarkers, particularly through the RAR. Zhao et al. (2021) reported a significant correlation between elevated RAR and mortality in patients with stroke admitted to intensive care units [20]. Eyiol et al. (2024) further revealed that patients with severe stroke presented substantially higher RAR levels than patients with non-severe stroke did (P<0.001), with strong associations with NIH Stroke Scale scores (P<0.001) [21]. Longitudinal findings by Liu et al. (2022) strengthened this evidence, showing that log-transformed RAR values positively correlated with 30-day all-cause mortality (OR=4.02), ICU mortality (OR=3.81), and in-hospital mortality (OR=3.31) (all P<0.0001) [22]. Nevertheless, existing research predominantly targets general or critically ill patients with stroke, leaving the prognostic value of RAR in patients with AMIS largely unexplored. Given the potential interplay between erythrocyte dynamics and systemic inflammation in stroke pathophysiology, we hypothesize that an elevated RAR is associated with an increased risk of unfavorable functional outcomes in patients with AMIS.

Therefore, this study aimed to evaluate the associations between the RAR and 3-month functional outcomes in patients with AMIS, as assessed by the modified Rankin Scale (mRS). By examining the association of the RAR with this specific stroke subtype, we aimed to gain insights into the potential role of this readily available and cost-effective marker in improving risk stratification and guiding clinical decision-making for patients with AMIS.

## Methods

### Study design

This was a cohort study based on a previously conducted single-center prospective study performed in South Korea between January 2010 and December 2016 [23].

Here, the RAR was utilized as an independent variable, whereas the mRS score for patients with AMIS served as the dependent variable.

## Data source

Kang et al. [23] kindly made the original data used to conduct this study freely available. The findings were released through an unrestricted access model with a Creative Commons Attribution license, allowing for their unlimited sharing, distribution, and reproduction in any form so long as the initial source is properly acknowledged. We thank all the parties who contributed the data used for this research effort.

## Study population

The primary study included 2,084 individuals diagnosed with AIS, of whom 178 were excluded based on exclusion criteria established in the original study [23], including (1) the lack of any evaluation of dysphagia or laboratory testing conducted within 24 h of admission (n = 72) and (2) the lack of mRS scores calculated 3 months following hospital discharge (n = 106). The remaining 1,906 patients were retained in the initial study dataset. For this secondary analysis, AMIS was defined as an NIHSS score ≤3, so patients with an NIHSS score greater than 3 were excluded (n = 918). The patient selection process is outlined in Fig 1.

Patients with AIS admitted to the hospital within 7 days of symptom onset were recruited by the original investigator team [23], who used data from a single-center prospective registry [23]. The primary study was approved by Seoul National University Hospital's institutional review board, which waived any requirement for informed patient consent (IRB No. 1009-062-332) [23]. No additional ethical oversight was deemed necessary for secondary analyses. The original study

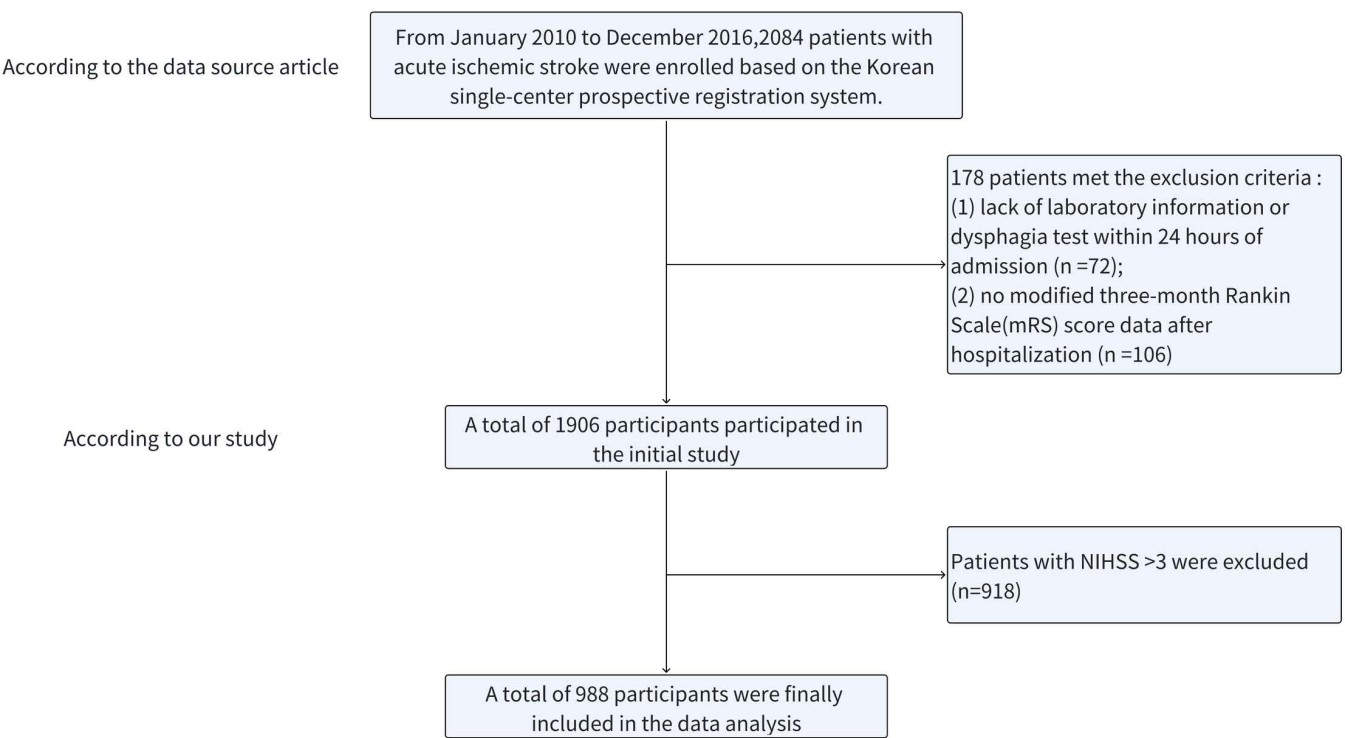

**Fig 1. An overview of the selection process.**

was performed as per the Declaration of Helsinki in accordance with appropriate laws and guidelines, and the same was true for this secondary analysis.

## Variables

RDW and ALB data measured within 24 hours of admission were obtained from electronic medical records [23], and the RAR was calculated as RDW (%)/albumin (g/dL).

## Patient 3-month prognostic outcomes

Patient outcomes at 3 months after AMIS onset were analyzed on the basis of mRS scores [24]. Follow-up information was collected via outpatient visits or telephone calls [23]. Patients were classified into those associated with a good prognosis (mRS ≤ 2) and those associated with a poor prognosis (mRS ≥ 3) [25].

## Covariates

All information related to data collection was derived from the original study [23]. Covariate selection was performed as per prior studies and our clinical experience. The analyzed covariates included the following: (1) continuous variables: body mass index (BMI), white blood cell count (WBC), red blood cell count (RBC), platelet count (PLT), erythrocyte Sedimentation Rate (ESR), total cholesterol (TC), triglyceride (TG), high-density lipoprotein (HDL), low-density lipoprotein (LDL), blood urea nitrogen (BUN), creatinine (CR), glycosylated hemoglobin (HbA1c), high-sensitivity C-reactive protein (hs-CRP), the international normalized ratio (INR), activated partial thromboplastin time (APTT), ALB, RDW, and NIHSS scores; and (2) categorical variables: age, sex, current smoking status, history of stroke or transient ischemic attack (TIA), hypertension, diabetes, hyperlipidemia, atrial fibrillation, coronary heart disease, and stroke etiology.

## Statistical analysis

The means and medians are presented for continuous variables, whereas frequencies and percentages are presented for categorical variables. First, we used the $\chi2$ test (for categorical variables), one-way analysis of variance (for normally distributed data) or the Kruskal–Wallis H test (for skewed data) to test the differences between different RAR levels (tertiles). Second, univariate logistic regression analysis was used to evaluate the impact of each variable on poor prognosis at 3 months after AMIS. Finally, three different multivariate binary logistic regression models were used to assess the relationship between the RAR and poor 3-month prognostic outcomes. In Model 1, no adjustment was made. In Model 2, sociodemographic variables were controlled for. In Model 3, a full adjustment was made for the variables, including age, gender, BMI, RBC, ESR, TC, HDL, hs-CRP, initial NIHSS, hypertension, diabetes, and stroke etiology. Confounding factors were assessed by evaluating the impact of covariates on the effect estimate of the RAR exceeding 10% or the regression coefficient of covariates on poor prognosis at 3 months with a significance level of P < 0.1 [26]. To assess the reliability of the results, sensitivity analyses were conducted. The RAR values were converted into tertiles as a categorical variable, and the trend P value was calculated to validate the results. Finally, subgroup analyses were performed, and subgroup interactions were examined with a likelihood ratio test. EmpowerStats (X&Y Solutions, Inc., MA) and R (http://www.r-project.org) were used for the statistical analyses, and a two-sided P < 0.05 was regarded as significant.

## Results

### Baseline characteristics of the study population

In this study, the fundamental characteristics of 988 participants were analyzed, and they were classified into low, medium and high groups based on the tertiles of the RAR levels (Table 1). The findings revealed a statistically significant relationship between increasing RAR levels and advancing age, particularly within the 70–80 age group, where the high RAR group presented a 40.61% prevalence (P < 0.001). Additionally, the study noted a correlation between BMI, WBC, and RBC

**Table 1. Baseline characteristics of the study population.**

| RDW/ALB tertile | Low(2.31–3.00) | Middle(3.00–3.32) | High(3.32–7.06) | P-value |
|---|---|---|---|---|
| No. of subjects | 325 | 333 | 330 | |
| Age, n (%) | | | | **<0.001** |
| < 60 | 124 (38.15%) | 74 (22.22%) | 60 (18.18%) | |
| 60 to < 70 | 93 (28.62%) | 104 (31.23%) | 77 (23.33%) | |
| 70 to < 80 | 89 (27.38%) | 114 (34.23%) | 134 (40.61%) | |
| ≥ 80 | 19 (5.85%) | 41 (12.31%) | 59 (17.88%) | |
| Gender, n (%) | | | | 0.561 |
| Male | 212 (65.23%) | 222 (66.67%) | 207 (62.73%) | |
| Female | 113 (34.77%) | 111 (33.33%) | 123 (37.27%) | |
| BMI, mean (SD), kg/m$^2$ | 24.49 (3.01) | 23.60 (2.92) | 23.25 (3.16) | **<0.001** |
| WBC, mean (SD), 10$^9$/L | 8.16 (2.67) | 7.44 (2.05) | 7.92 (3.16) | **0.002** |
| RBC, mean (SD), 10$^{12}$/L | 4.64 (0.48) | 4.41 (0.53) | 4.08 (0.67) | **<0.001** |
| PLT, mean (SD), 10$^9$/L | 238.78 (65.77) | 222.15 (53.29) | 218.90 (76.73) | **<0.001** |
| ESR, median (min-max), mm/h | 0.00 (0.00-100.00) | 0.00 (0.00-51.00) | 0.00 (0.00-112.00) | **0.007** |
| TC, mean (SD), mg/dL | 195.40 (40.03) | 179.22 (43.04) | 170.81 (39.68) | **<0.001** |
| TG, mean (SD), mg/dL | 120.53 (67.40) | 105.54 (61.82) | 101.32 (54.17) | **<0.001** |
| HDL, mean (SD), mg/dL | 45.66 (16.16) | 44.64 (16.66) | 42.50 (16.33) | **0.042** |
| LDL, mean (SD), mg/dL | 113.95 (40.17) | 101.62 (41.07) | 97.89 (40.74) | **<0.001** |
| BUN, mean (SD), mg/dL | 15.77 (6.51) | 16.55 (6.35) | 18.88 (10.40) | **<0.001** |
| CR, median (min-max), mg/dL | 0.88 (0.40-9.86) | 0.91 (0.36-11.12) | 0.90 (0.40-13.91) | **0.038** |
| HbA1c, mean (SD), % | 4.92 (2.84) | 5.01 (2.57) | 5.26 (2.72) | 0.264 |
| hs-CRP, median (min-max), mg/dL | 0.07 (0.00-3.25) | 0.08 (0.00-14.92) | 0.18 (0.00-30.81) | **<0.001** |
| INR, median (min-max) | 0.97 (0.00-4.47) | 0.97 (0.00-2.70) | 0.99 (0.00-7.47) | **<0.001** |
| APTT, mean (SD), s | 31.23 (4.96) | 31.20 (6.96) | 31.13 (7.52) | 0.981 |
| Initial NIHSS score, median (min-max) | 1.00 (0.00-3.00) | 1.00 (0.00-3.00) | 2.00 (0.00-3.00) | 0.487 |
| RDW, mean (SD), % | 12.47 (0.52) | 13.00 (0.56) | 14.28 (1.84) | **<0.001** |
| ALB, mean (SD), g/dL | 4.43 (0.19) | 4.13 (0.18) | 3.74 (0.33) | **<0.001** |
| Current smoking, n (%) | | | | 0.114 |
| No | 177 (54.46%) | 208 (62.46%) | 194 (58.79%) | |
| Yes | 148 (45.54%) | 125 (37.54%) | 136 (41.21%) | |
| Previous stroke/TIA, n (%) | | | | 0.275 |
| No | 277 (85.23%) | 269 (80.78%) | 269 (81.52%) | |
| Yes | 48 (14.77%) | 64 (19.22%) | 61 (18.48%) | |
| Hypertension, n (%) | | | | 0.388 |
| No | 125 (38.46%) | 118 (35.44%) | 110 (33.33%) | |
| Yes | 200 (61.54%) | 215 (64.56%) | 220 (66.67%) | |
| Diabetes, n (%) | | | | 0.107 |
| No | 236 (72.62%) | 237 (71.17%) | 216 (65.45%) | |
| Yes | 89 (27.38%) | 96 (28.83%) | 114 (34.55%) | |
| Hyperlipidemia, n (%) | | | | **0.009** |
| No | 181 (55.69%) | 207 (62.16%) | 222 (67.27%) | |
| Yes | 144 (44.31%) | 126 (37.84%) | 108 (32.73%) | |
| Atrial fibrillation, n (%) | | | | **0.037** |
| No | 291 (89.54%) | 277 (83.18%) | 276 (83.64%) | |
| Yes | 34 (10.46%) | 56 (16.82%) | 54 (16.36%) | |

*(Continued)*

**Table 1.** (Continued)

| RDW/ALB tertile | Low(2.31–3.00) | Middle(3.00–3.32) | High(3.32–7.06) | P-value |
|---|---|---|---|---|
| Coronary heart disease, n (%) | | | | 0.063 |
| No | 301 (92.62%) | 294 (88.29%) | 288 (87.27%) | |
| Yes | 24 (7.38%) | 39 (11.71%) | 42 (12.73%) | |
| mRS score, n (%) | | | | **<0.001** |
| 0–2 | 300 (92.31%) | 307 (92.19%) | 272 (82.42%) | |
| 3–6 | 25 (7.69%) | 26 (7.81%) | 58 (17.58%) | |
| Stroke etiology, n (%) | | | | **<0.001** |
| SVO | 112 (34.57%) | 101 (30.33%) | 103 (31.21%) | |
| LAA | 91 (28.09%) | 95 (28.53%) | 60 (18.18%) | |
| CE | 48 (14.81%) | 77 (23.12%) | 78 (23.64%) | |
| Other determined | 19 (5.86%) | 16 (4.80%) | 46 (13.94%) | |
| Undetermined | 54 (16.67%) | 44 (13.21%) | 43 (13.03%) | |

Abbreviations: BMI, body mass index; WBC, white blood cell; RBC, red blood cell; PLT, platelet count; ESR, erythrocyte sedimentation rate; TC, total cholesterol; TG, triglycerides; HDL, high-density lipoproteins; LDL, low-density lipoproteins; BUN, blood Urea Nitrogen; CR, creatinine; HbA1c, glycosylated hemoglobin; hs-CRP, high sensitive c-reactive protein; INR, international normalized ratio; APTT, activated partial thromboplastin time; NIHSS, National Institutes of Health Stroke Scale; RDW, red blood cell distribution width; ALB, albumin; TIA, transient ischemia attack; mRS, modified rankin scale; SVO, small-vessel occlusion; LAA, large-artery atherosclerosis; CE, cardioembolism.

parameters. RBC and PLT counts were significantly lower in the high group (P<0.001), whereas TC, TG and LDL levels were also significantly lower (P<0.001). With respect to clinical characteristics, the incidence of hyperlipidemia and atrial fibrillation was significantly lower in the high group (P=0.009; P=0.037), whereas the proportion of patients with an mRS score ≥3 was significantly greater (P<0.001). The analysis of stroke etiology revealed that the proportion of patients with small-vessel occlusion (SVO) was highest in the low group, whereas the incidence of cardiogenic embolism (CE) was increased in the high group (P<0.001).

## Univariate analysis of 3-month adverse outcomes in AMIS patients

Univariate analysis revealed several factors significantly associated with poor prognosis at 3 months (Table 2). These included lower RBC counts, higher ESRs, higher hs-CRP levels, higher initial NIHSS scores, higher RDWs, lower ALB levels, higher RARs, and hypertension. In terms of stroke etiology, patients with CE stroke and undetermined etiology had a lower risk of poor prognosis than those with SVO.

## Association between RAR and poor 3-month prognosis in patients with AMIS

As shown in Table 3, according to the unadjusted model, the RAR ratio was significantly associated with poor prognosis (OR=2.29, 95% CI: 1.74–3.02; P<0.0001). This association remained significant after adjusting for age and sex (OR=2.18, 95% CI: 1.63–2.91; P<0.0001). Further adjustment for additional variables strengthened the association (OR=2.44, 95% CI: 1.67–3.57, P<0.0001). Additionally, tertile analysis of the RAR revealed that the high tertile group was significantly associated with poor prognosis (Model 1: OR=2.56, P=0.0002; Model 2: OR=1.99, P=0.0092; Model 3: OR=1.86, P=0.0400), whereas the middle tertile group showed no significant association. Trend tests further confirmed a significant trend between the RAR and poor prognosis (P for trend <0.0001, 0.0033, 0.0259).

## Subgroup analysis of the relationship between the RAR and 3-month poor prognosis in patients with AMIS

As detailed in Table 4, stratified analysis revealed that although significant interactions were detected in some subgroups (such as sex, hypertension, diabetes, and hyperlipidemia, the P values for interaction were 0.0214, 0.0081, 0.0368, and

**Table 2. Univariate analysis of 3-month adverse outcomes in AMIS patients.**

| | Statistics | Odds ratio (95% CI) | P-value |
|---|---|---|---|
| Age, n (%) | | | |
| < 60 | 258 (26.11%) | 1.0 | |
| 60 to < 70 | 274 (27.73%) | 0.78 (0.39, 1.55) | 0.4792 |
| 70 to < 80 | 337 (34.11%) | 1.84 (1.04, 3.24) | **0.0348** |
| ≥ 80 | 119 (12.04%) | 4.43 (2.38, 8.25) | **<0.0001** |
| Gender, n (%) | | | |
| Male | 641 (64.88%) | 1.0 | |
| Female | 347 (35.12%) | 1.23 (0.82, 1.85) | 0.3162 |
| BMI, mean (SD), kg/m² | 23.77±3.07 | 0.91 (0.85, 0.97) | **0.0041** |
| WBC, mean (SD), 10⁹/L | 7.84±2.68 | 1.05 (0.98, 1.13) | 0.1469 |
| RBC, mean (SD), 10¹²/L | 4.38±0.61 | 0.64 (0.46, 0.87) | **0.0052** |
| PLT, mean (SD), 10⁹/L | 226.53±66.43 | 1.00 (1.00, 1.00) | 0.2041 |
| ESR, median (min-max), mm/h | 0.00 (0.00-112.00) | 1.02 (1.01, 1.03) | **0.0003** |
| TC, mean (SD), mg/dL | 181.73±42.16 | 1.00 (1.00, 1.01) | 0.5908 |
| TG, mean (SD), mg/dL | 109.07±61.83 | 1.00 (1.00, 1.00) | 0.4467 |
| HDL, mean (SD), mg/dL | 44.26±16.42 | 0.99 (0.98, 1.00) | 0.0591 |
| LDL, mean (SD), mg/dL | 104.43±41.19 | 1.00 (1.00, 1.01) | 0.8069 |
| BUN, mean (SD), mg/dL | 17.07±8.08 | 1.01 (0.99, 1.04) | 0.228 |
| CR, median (min-max), mg/dL | 0.89 (0.36-13.91) | 1.03 (0.88, 1.21) | 0.6795 |
| HbA1c, mean (SD), % | 5.07±2.71 | 1.02 (0.95, 1.10) | 0.6295 |
| hs-CRP, median (min-max), mg/dL | 0.09 (0.00-30.81) | 1.08 (1.02, 1.15) | **0.0061** |
| INR, median (min-max) | 0.98 (0.00-7.47) | 1.38 (0.93, 2.05) | 0.1116 |
| APTT, mean (SD), s | 31.19±6.57 | 1.01 (0.99, 1.04) | 0.3567 |
| Initial NIHSS score, median (min-max) | 1.00 (0.00-3.00) | 1.49 (1.23, 1.81) | **<0.0001** |
| RDW, mean (SD), % | 13.26±1.38 | 1.28 (1.15, 1.44) | **<0.0001** |
| ALB, mean (SD), g/dL | 4.10±0.37 | 0.30 (0.19, 0.50) | **<0.0001** |
| RDW/ALB | 3.28±0.57 | 2.29 (1.74, 3.02) | **<0.0001** |
| Current smoking, n (%) | | | |
| No | 579 (58.60%) | 1 | |
| Yes | 409 (41.40%) | 0.80 (0.53, 1.21) | 0.2916 |
| Previous stroke, n (%) | | | |
| No | 815 (82.49%) | 1 | |
| Yes | 173 (17.51%) | 1.38 (0.85, 2.25) | 0.1907 |
| Hypertension, n (%) | | | |
| No | 353 (35.73%) | 1 | |
| Yes | 635 (64.27%) | 2.00 (1.25, 3.19) | **0.0036** |
| Diabetes, n (%) | | | |
| No | 689 (69.74%) | 1 | |
| Yes | 299 (30.26%) | 1.45 (0.96, 2.20) | 0.0777 |
| Hyperlipidemia, n (%) | | | |
| No | 610 (61.74%) | 1 | |
| Yes | 378 (38.26%) | 0.97 (0.64, 1.46) | 0.8833 |
| Atrial fibrillation, n (%) | | | |
| No | 844 (85.43%) | 1 | |
| Yes | 144 (14.57%) | 0.77 (0.42, 1.42) | 0.4072 |

*(Continued)*

**Table 2.** (Continued)

| | Statistics | Odds ratio (95% CI) | P-value |
|---|---|---|---|
| Coronary heart disease, n (%) | | | |
| No | 883 (89.37%) | 1 | |
| Yes | 105 (10.63%) | 0.55 (0.25, 1.21) | 0.1364 |
| Stroke etiology, n (%) | | | |
| SVO | 316 (32.02%) | 1 | |
| LAA | 246 (24.92%) | 0.67 (0.39, 1.13) | 0.1347 |
| CE | 203 (20.57%) | 0.53 (0.29, 0.97) | **0.038** |
| Other determined | 81 (8.21%) | 1.52 (0.81, 2.87) | 0.1936 |
| Undetermined | 141 (14.29%) | 0.37 (0.17, 0.81) | **0.0131** |

**Table 3.** Association between RAR and poor 3-month prognosis in patients with AMIS in different models.

| Exposure | Model 1 OR (95%CI) P value | Model 2 OR (95%CI) P value | Model 3 OR (95%CI) P value |
|---|---|---|---|
| RDW/ALB | 2.29 (1.74, 3.02) <0.0001 | 2.18 (1.63, 2.91) <0.0001 | 2.44 (1.67, 3.57) <0.0001 |
| RDW/ALB tertile | | | |
| Low | Ref | Ref | Ref |
| Middle | 1.02 (0.57, 1.80) 0.9558 | 0.88 (0.49, 1.57) 0.6565 | 0.95 (0.51, 1.77) 0.8761 |
| High | 2.56 (1.56, 4.21) 0.0002 | 1.99 (1.19, 3.34) 0.0092 | 1.86 (1.03, 3.35) 0.0400 |
| P for trend | 1.69 (1.30, 2.19) <0.0001 | 1.49 (1.14, 1.95) 0.0033 | 1.41 (1.04, 1.90) 0.0259 |

Model 1: Non-adjusted model.

Model 2: Adjusted for Age and Gender.

Model 3: Adjusted for Age, Gender, BMI, RBC, ESR, TC, HDL, hs-CRP, Initial NIHSS, Hypertension, Diabetes, Stroke etiology.

0.0081, respectively), the associations between the RAR and the 3-month poor prognosis of acute minor ischemic stroke remained consistent across all subgroups. Specifically, RDW/ALB was positively correlated with poor prognosis in all subgroups, and the odds ratio (OR) was greater than 1. The results of the stratified analysis of other variables (such as age, current smoking status, previous stroke/TIA, atrial fibrillation, coronary heart disease and stroke etiology) further supported this consistency. Although the correlation of some subgroups was not statistically significant (P > 0.05), the OR values were all greater than 1, indicating that the correlation between RDW/ALB and poor prognosis did not change.

## Discussion

To our knowledge, this is the first study to investigate the association between the RAR and 3-month poor prognosis (mRS ≥ 3) in patients with AMIS. Our study revealed a positive linear association between the RAR and 3-month poor prognosis in patients with AMIS. This association remained robust after adjustment for potential confounders (OR=2.44, 95% CI: 1.67–3.57). Stratified analyses further revealed that the positive association between the RAR and poor prognosis was consistent across subgroups (all ORs > 1), despite subgroup interactions for sex and hypertension (P interaction <0.05).

Recent studies consistently indicate that the RAR holds significant clinical value in prognostic assessment across various diseases. For example, a joint analysis based on the MIMIC-III database and a heart failure cohort from Wenzhou Medical University revealed a stepwise increase in the 90-day mortality risk for the medium-high RAR groups (4.33–5.44 and >5.44) compared with the lowest RAR tertile group (RAR < 4.33), with hazard ratios (HRs) of 2.00 and

**Table 4. Subgroup analysis of the relationship between the RAR and 3-month poor prognosis in patients with AMIS.**

| Subgroup | No. of subjects | Odds ratio (95% CI) | P-value | P for interaction |
|---|---|---|---|---|
| Age, year | | | | 0.6269 |
| < 60 | 258 | 3.48 (1.47, 8.23) | 0.0046 | |
| 60 to < 70 | 274 | 3.85 (1.72, 8.63) | 0.001 | |
| 70 to < 80 | 337 | 2.20 (1.16, 4.15) | 0.0154 | |
| ≥ 80 | 119 | 2.26 (0.98, 5.20) | 0.0556 | |
| Gender | | | | **0.0214** |
| Male | 641 | 3.56 (2.12, 5.98) | <0.0001 | |
| Female | 347 | 1.42 (0.77, 2.62) | 0.2579 | |
| Current smoking | | | | 0.4630 |
| No | 579 | 2.62 (1.65, 4.18) | <0.0001 | |
| Yes | 409 | 1.94 (1.00, 3.77) | 0.0513 | |
| Previous stroke/TIA | | | | 0.9259 |
| No | 815 | 2.69 (1.70, 4.23) | <0.0001 | |
| Yes | 173 | 2.56 (1.03, 6.37) | 0.0437 | |
| Hypertension | | | | **0.0081** |
| No | 353 | 6.19 (2.84, 13.49) | <0.0001 | |
| Yes | 635 | 2.03 (1.33, 3.11) | 0.0010 | |
| Diabetes | | | | **0.0368** |
| No | 689 | 1.81 (1.10, 2.98) | 0.0206 | |
| Yes | 299 | 4.11 (2.22, 7.60) | <0.0001 | |
| Hyperlipidemia | | | | **0.0081** |
| No | 610 | 6.19 (2.84, 13.49) | <0.0001 | |
| Yes | 378 | 2.03 (1.33, 3.11) | 0.0010 | |
| Atrial fibrillation | | | | 0.6164 |
| No | 844 | 2.50 (1.62, 3.88) | <0.0001 | |
| Yes | 144 | 3.67 (0.85, 15.80) | 0.0811 | |
| Coronary heart disease | | | | 0.7673 |
| No | 883 | 2.84 (1.86, 4.35) | <0.0001 | |
| Yes | 105 | 3.70 (0.68, 20.19) | 0.1311 | |
| Stroke etiology | | | | 0.1111 |
| SVO | 316 | 4.29 (1.94, 9.48) | 0.0003 | |
| LAA | 246 | 1.37 (0.48, 3.87) | 0.5551 | |
| CE | 203 | 2.32 (0.87, 6.15) | 0.0915 | |
| Other determined | 81 | 2.14 (0.83, 5.52) | 0.1147 | |
| Undetermined | 141 | 51.57 (1.96, 1355.64) | 0.0181 | |

Notes: Above model adjusted for Age, Gender, BMI, RBC, ESR, TC, HDL, hs-CRP, Initial NIHSS, Hypertension, Diabetes, and Stroke etiology. In each case, the model is not adjusted for the stratification variable.

3.63, respectively. Even after adjusting for confounding factors such as age, sex, and the SOFA score, the high RAR group maintained independent predictive power (adjusted HR = 2.70) [27]. This finding aligns with the prognostic patterns observed in patients with acute myocardial infarction: a multivariate analysis of 2,594 cases from the eICU database confirmed that each unit increase in the RAR was associated with a 27% increase in in-hospital mortality risk (OR=1.27, 95% CI 1.12–1.43), and the survival curve for the high RAR group (≥4.776) was significantly lower than that of the low RAR group (P<0.0001) [28]. Notably, this association is also significant in chronic diseases. A long-term follow-up study of

997 patients with CKD in the Fukushima cohort revealed that the risk of progression to end-stage kidney disease (ESKD) in the medium–high RAR groups (T2-T3) was increased by a gradient of 1.372.92 times compared with that in the lowest baseline RAR tertile group (T1) [29]. Furthermore, an analysis of MIMIC-III data focused on patients with sepsis extended the clinical applicability of the RAR: RAR levels in nonsurvivors were significantly higher than those in survivors at both the 30-day and 90-day endpoints, and an elevated RAR was significantly associated with mortality risk in a dose–response manner [30].

Although previous studies have confirmed the prognostic value of the RAR in various diseases, relatively few studies have focused on the relationship between the RAR and adverse stroke outcomes. Existing research suggests that the RAR may play an important role in the prognostic assessment of patients with stroke. A study based on the MIMIC-III database, which included 1480 patients with stroke, revealed that higher RAR values were significantly associated with mortality in patients with stroke in the ICU. After adjusting for age and sex, the 30-day, 90-day, and one-year all-cause mortality rates were significantly higher in the high RAR group than in the reference group (HRs were 1.88, 2.12, and 2.15, respectively) [20]. Another retrospective cohort analysis of 1412 patients with acute ischemic stroke (AIS) from the MIMIC-IV database revealed a significant positive correlation between the log-transformed RAR value and the risk of death. Specifically, higher RAR values were significantly associated with increased 30-day all-cause mortality (OR = 4.02), ICU mortality (OR = 3.81), and in-hospital mortality (OR = 3.31) (all $P < 0.0001$) (Liu et al.,2022). Furthermore, a prospective study of 127 patients with AIS in which the NIHSS score was used to assess disease severity revealed that RAR levels were significantly higher in severely ill patients than in nonseverely ill patients ($p < 0.001$), and the RAR was significantly positively correlated with NIHSS scores ($p < 0.001$), suggesting that the RAR may reflect the severity of stroke [21]. Consistent with the above studies, our study further confirms the role of the RAR in stroke prognosis. We focused on patients with AMIS and used the 3-month mRS score as a prognostic indicator. The results revealed a significant positive linear correlation between the RAR and poor 3-month prognosis (mRS ≥ 3) in AMIS patients; i.e., for every one-unit increase in the RAR, the risk of poor 3-month prognosis increased by 1.44 times. In summary, our study expands the existing literature and further highlights the potential of the RAR as an important biomarker in the prognostic assessment of AMIS patients with AMIS.

Considering the potential mechanisms by which the RAR may contribute to unfavorable outcomes in patients with AMIS, the extant evidence suggests the following pathways as potential points of convergence: 1) Inflammation-oxidative stress axis: Elevated RDWs are associated with increased levels of proinflammatory factors (e.g., IL-6, CRP) and reactive oxygen species (ROS) [29,31–34]. Conversely, ALB has been shown to have anti-inflammatory properties by scavenging free radicals and inhibiting the nuclear factor-κB (NF-κB) pathway [29,35,36]. An elevated RAR may indicate an imbalance between inflammation and antioxidant capacity, potentially exacerbating ischemic penumbra injury. 2) Endothelial dysfunction: ALB protects endothelial function by maintaining nitric oxide (NO) bioavailability [29,37,38], whereas elevated RDW may reflect decreased endothelial repair capacity, promoting thrombosis and impaired collateral circulation [29,39,40]. 3) Blood rheology abnormalities: Elevated RDW leads to decreased erythrocyte deformability and increased microcirculatory resistance [29,6], whereas low ALB levels aggravate cerebral edema by decreasing plasma colloid osmolality [29,41]. The synergistic effect of these two factors may exacerbate cerebral perfusion, leading to increased neurological deficits.

The RAR, derived from routine blood tests (RDW and albumin), offers a cost-effective tool for early risk stratification in AMIS. Elevated RAR levels could identify high-risk patients who may benefit from intensified monitoring or aggressive secondary prevention (e.g., dual antiplatelet therapy), even with mild initial symptoms. Future studies should explore whether RAR-guided interventions improve long-term outcomes.

The present study has several merits. First, to the best of our knowledge, this is the first study to explore the relationship between the RAR and poor prognosis in patients with AMIS. Second, the independent association between the RAR and 3-month poor prognosis (mRS ≥ 3) was robustly validated using a combination of univariate analysis, multifactorial stepwise-adjusted logistic regression modeling, and sensitivity analysis. Third, the clinical application potential of the RAR

as an easily accessible and low-cost test in routine blood tests provides a practical tool for risk stratification of patients with AMIS.

However, this study has several limitations. First, the analyses were conducted in a cohort of Korean patients; therefore, validation in other ethnic groups is necessary. Second, in the original study, age was considered a categorical variable rather than a continuous variable. Consequently, the available data may be incomplete. Additionally, the RAR was measured only once at admission, precluding the assessment of the relationship between dynamic RAR changes and poor prognosis after stroke. Finally, this was a secondary analysis of a published dataset, and the original data did not include details related to potentially relevant confounders, including detailed medication history or intravenous thrombolysis.

## Conclusion

The results of this study demonstrated a linear and positive correlation between the RAR and three-month adverse prognosis in patients with AMIS in the Korean population. The RAR offers a practical tool for early risk stratification and individualized management of patients with AMIS.

## Supporting information

**S1 Data.  Minimum dataset for all analyses.**
(XLS)

## Acknowledgments

As a secondary study, the data and explanation of the methodology are mainly taken from: Kim Y, Nam K-W, Jeong H-Y, Kang MK, Kim TJ, Kim SK, et al. (2020) Automated undernutrition screen tool: Geriatric nutritional risk score predicts poor outcomes in individuals with acute ischemic stroke. https://doi.org/10.1371/journal.pone. 0228738; PLoS ONE 15(2): e0228738. All authors of the study deserve our appreciation.

## Author contributions

**Conceptualization:** Lei Xu.

**Data curation:** Qin Chen, Qin Xiong, Lei Xu.

**Formal analysis:** Qin Chen, Qin Xiong, Li Liu, Lei Xu.

**Writing – original draft:** Qin Chen.

**Writing – review & editing:** Qin Chen, Qin Xiong, Li Liu, Lei Xu.

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
