## [Decision Letter · Decision Letter 0]

PONE-D-25-24758Associations between the red blood cell distribution width-to-albumin ratio and 3-month outcomes in patients with acute minor ischemic stroke: A cohort studyPLOS ONE

Dear Dr. Xu,

"Associations between the red blood cell distribution width-to-albumin ratio and 3-month outcomes in patients with acute minor ischemic stroke: A cohort study " stands out as a nice compilation. However, some editing will be needed.

It is recommended that a process of re-evaluation be undertaken following each revision.

We look forward to receiving your revised manuscript.

Kind regards,

Ufuk Demirci, MD

Academic Editor

PLOS ONE

Additional Editor Comments:

Associations between the red blood cell distribution width-to-albumin ratio and 3-month outcomes in patients with acute minor ischemic stroke: A cohort study

Reviewers' comments:

Reviewer's Responses to Questions

**Comments to the Author**

1. Is the manuscript technically sound, and do the data support the conclusions?

Reviewer #1: Yes

Reviewer #2: Yes

2. Has the statistical analysis been performed appropriately and rigorously? 

Reviewer #1: Yes

Reviewer #2: I Don't Know

3. Have the authors made all data underlying the findings in their manuscript fully available?

Reviewer #1: Yes

Reviewer #2: Yes

4. Is the manuscript presented in an intelligible fashion and written in standard English?

Reviewer #1: Yes

Reviewer #2: Yes

5. Review Comments to the Author

Reviewer #1: The study looks at the association between red cell distribution width-to-albumin ratio and three month mortality among acute minor ischemic stroke patients.

The article is very well written and features sound statistical analysis. I would recommend the following minor corrections:

Line 172: Please consider elaborating "SVO" as small-vessel occlusion upon first mention in the text. While it is defined in the accompanying table, introducing the full term at first use in the main text would improve clarity.

Line 274: The term "hemidiaphragm injury" appears to be an error. Please review this reference, as it may be a misstatement or require clarification.

Reviewer #2: This is a well-conducted and clearly presented cohort study examining the association between the red blood cell distribution width-to-albumin ratio (RA) and 3-month outcomes in patients with acute minor ischemic stroke. The methodology is appropriate, and the statistical analyses are sound. The findings contribute meaningfully to the growing literature on prognostic biomarkers in stroke.

I have no major concerns and recommend acceptance of the manuscript. However, it would strengthen the discussion if the authors could briefly elaborate on how their findings might be translated into clinical care or influence clinical decision-making. For instance, how might RA be used in risk stratification or management of patients with minor stroke?

6. PLOS authors have the option to publish the peer review history of their article (what does this mean? ). If published, this will include your full peer review and any attached files.

**Do you want your identity to be public for this peer review?** For information about this choice, including consent withdrawal, please see our Privacy Policy .

Reviewer #1: No

Reviewer #2: No

---

## [Author Response · Author response to Decision Letter 1]

16 Jun 2025

Dear Dr. Ufuk Demirci and Reviewers,

We sincerely appreciate your thoughtful feedback on our manuscript entitled "Associations between the red blood cell distribution width-to-albumin ratio and 3-month outcomes in patients with acute minor ischemic stroke: A cohort study". The insights provided by you and the reviewers have been invaluable in guiding us to improve the manuscript. We have carefully considered each suggestion and strived to implement comprehensive revisions, hoping these adjustments align with the journal’s standards and your expectations. Revised sections are highlighted with red track changes in the manuscript. Below is a summary of the key modifications and our responses to the reviewers’ comments:

Response: We sincerely appreciate your invaluable suggestions provided during the review process. These suggestions have been instrumental in enabling us to enhance the manuscript to a greater extent. Concerning the format requirements of the journal, we have carried out a thorough and meticulous check, and subsequently made all the necessary revisions to ensure full compliance.

2.PLOS requires an ORCID iD for the corresponding author in Editorial Manager on papers submitted after December 6th, 2016. Please ensure that you have an ORCID iD and that it is validated in Editorial Manager. To do this, go to ‘Update my Information’ (in the upper left-hand corner of the main menu), and click on the Fetch/Validate link next to the ORCID field. This will take you to the ORCID site and allow you to create a new iD or authenticate a pre-existing iD in Editorial Manager.

Response: We sincerely appreciate your patient reminder and detailed guidance! We have now completed the update and validation of the ORCID iD as required. If there are any oversights or areas for improvement in the subsequent process, please kindly let us know. We will fully cooperate to rectify and perfect them.

Response: We have confirmed the ethics statement is exclusively retained in the Methods section and removed from all other sections as required.

Response: Thank you for your guidance regarding the reference list. We have carefully and thoroughly reviewed each citation in the manuscript, striving to ensure its completeness and accuracy. To the best of our knowledge, no retracted papers were identified among the references cited.

Reviewer #1: The study looks at the association between red cell distribution width-to-albumin ratio and three month mortality among acute minor ischemic stroke patients.

The article is very well written and features sound statistical analysis. I would recommend the following minor corrections:

Line 172: Please consider elaborating "SVO" as small-vessel occlusion upon first mention in the text. While it is defined in the accompanying table, introducing the full term at first use in the main text would improve clarity.

Response: Thank you for your valuable suggestion. At Line 166, when making reference to 'SVO', we have already presented the complete term 'small-vessel occlusion'. This measure has been taken to improve the readability of the main text, ensuring that readers can easily understand the context without ambiguity.

Line 274: The term "hemidiaphragm injury" appears to be an error. Please review this reference, as it may be a misstatement or require clarification.

Response: Thank you for catching this error. We corrected "hemidiaphragm injury" to "ischemic penumbra" at Line 269, aligning with standard terminology. Much appreciated for your careful review!

Reviewer #2: This is a well-conducted and clearly presented cohort study examining the association between the red blood cell distribution width-to-albumin ratio (RA) and 3-month outcomes in patients with acute minor ischemic stroke. The methodology is appropriate, and the statistical analyses are sound. The findings contribute meaningfully to the growing literature on prognostic biomarkers in stroke.

I have no major concerns and recommend acceptance of the manuscript. However, it would strengthen the discussion if the authors could briefly elaborate on how their findings might be translated into clinical care or influence clinical decision-making. For instance, how might RA be used in risk stratification or management of patients with minor stroke?

Response: Thank you for the suggestion. We added a paragraph in the Discussion (Lines 278-283) to discuss the RA ratio’s clinical implications in risk stratification and minor stroke management. Appreciate your insight!

---

## [Decision Letter · Decision Letter 1]

Associations between the red blood cell distribution width-to-albumin ratio and 3-month outcomes in patients with acute minor ischemic stroke: A cohort study

PONE-D-25-24758R1

Dear Dr. Lei Xu,

We’re pleased to inform you that your manuscript has been judged scientifically suitable for publication and will be formally accepted for publication once it meets all outstanding technical requirements.

Kind regards,

Ufuk Demirci, MD

Academic Editor

PLOS ONE

Additional Editor Comments (optional):

Dear Author,

"Associations between the red blood cell distribution width-to-albumin ratio and 3-month outcomes in patients with acute minor ischemic stroke: A cohort study " stands out as a nice compilation. Thank you for the revision. This is a study that can contribute to the literature.

Yours faithfully,

Reviewers' comments:

Reviewer's Responses to Questions

**Comments to the Author**

1. If the authors have adequately addressed your comments raised in a previous round of review and you feel that this manuscript is now acceptable for publication, you may indicate that here to bypass the “Comments to the Author” section, enter your conflict of interest statement in the “Confidential to Editor” section, and submit your "Accept" recommendation.

Reviewer #1: All comments have been addressed

2. Is the manuscript technically sound, and do the data support the conclusions?

Reviewer #1: Yes

3. Has the statistical analysis been performed appropriately and rigorously? 

Reviewer #1: Yes

4. Have the authors made all data underlying the findings in their manuscript fully available?

Reviewer #1: Yes

5. Is the manuscript presented in an intelligible fashion and written in standard English?

Reviewer #1: Yes

6. Review Comments to the Author

Reviewer #1: (No Response)

7. PLOS authors have the option to publish the peer review history of their article (what does this mean? ). If published, this will include your full peer review and any attached files.

**Do you want your identity to be public for this peer review?** For information about this choice, including consent withdrawal, please see our Privacy Policy .

Reviewer #1: No

---

## [Editor Report · Acceptance letter]

PONE-D-25-24758R1

PLOS ONE

Dear Dr. Xu,

I'm pleased to inform you that your manuscript has been deemed suitable for publication in PLOS ONE. Congratulations! Your manuscript is now being handed over to our production team.

Kind regards,

on behalf of

Dr. Ufuk Demirci

Academic Editor

PLOS ONE